# High SOX9 Maintains Glioma Stem Cell Activity through a Regulatory Loop Involving STAT3 and PML

**DOI:** 10.3390/ijms23094511

**Published:** 2022-04-19

**Authors:** Paula Aldaz, Natalia Martín-Martín, Ander Saenz-Antoñanzas, Estefania Carrasco-Garcia, María Álvarez-Satta, Alejandro Elúa-Pinin, Steven M. Pollard, Charles H. Lawrie, Manuel Moreno-Valladares, Nicolás Samprón, Jürgen Hench, Robin Lovell-Badge, Arkaitz Carracedo, Ander Matheu

**Affiliations:** 1Group of Cellular Oncology, Biodonostia Health Research Institute, 20014 San Sebastian, Spain; paula.aldaz.donamaria@navarra.es (P.A.); ander.saenz@biodonostia.org (A.S.-A.); estefania.carrasco@biodonostia.org (E.C.-G.); maria.alvarez@biodonostia.org (M.Á.-S.); morevall40@gmail.com (M.M.-V.); nicolas.sampron@me.com (N.S.); 2Center for Cooperative Research in Biosciences (CIC bioGUNE), Basque Research and Technology Alliance (BRTA), Bizkaia Technology Park, 48160 Derio, Spain; nmartin@cicbiogune.es (N.M.-M.); acarracedo@cicbiogune.es (A.C.); 3CIBER of Frailty and Healthy Aging (CIBERFES), Carlos III Institute, 28029 Madrid, Spain; 4Donostia University Hospital, 20014 San Sebastian, Spain; alejandro.elua.pinin@gmail.com; 5Centre for Regenerative Medicine & Edinburgh Cancer Research UK Centre, Institute for Regeneration and Repair, Edinburgh EH16 4UU, UK; steven.pollard@ed.ac.uk; 6Group of Molecular Oncology, Biodonostia Health Research Institute, 20014 San Sebastian, Spain; charles.lawrie@biodonostia.org; 7Ikerbasque, Basque Foundation for Science, 48009 Bilbao, Spain; 8Institute of Pathology, University Hospital Basel, 48009 Basel, Switzerland; juergen.hench@usb.ch; 9The Francis Crick Institute, London NW1 1AT, UK; robin.lovell-badge@crick.ac.uk; 10Biochemistry and Molecular Biology Department, University of the Basque Country (UPV/EHU), 48940 Leioa, Spain; 11CIBER of Cancer (CIBERONC), Carlos III Institute, 28029 Madrid, Spain

**Keywords:** glioblastoma, glioma stem cell, SOX9, therapy, transcriptome, STAT3, PML, pharmacological inhibition

## Abstract

Glioma stem cells (GSCs) are critical targets for glioma therapy. SOX9 is a transcription factor with critical roles during neurodevelopment, particularly within neural stem cells. Previous studies showed that high levels of SOX9 are associated with poor glioma patient survival. SOX9 knockdown impairs GSCs proliferation, confirming its potential as a target for glioma therapy. In this study, we characterized the function of SOX9 directly in patient-derived glioma stem cells. Notably, transcriptome analysis of GSCs with SOX9 knockdown revealed STAT3 and PML as downstream targets. Functional studies demonstrated that SOX9, STAT3, and PML form a regulatory loop that is key for GSC activity and self-renewal. Analysis of glioma clinical biopsies confirmed a positive correlation between SOX9/STAT3/PML and poor patient survival among the cases with the highest SOX9 expression levels. Importantly, direct STAT3 or PML inhibitors reduced the expression of SOX9, STAT3, and PML proteins, which significantly reduced GSCs tumorigenicity. In summary, our study reveals a novel role for SOX9 upstream of STAT3, as a GSC pathway regulator, and presents pharmacological inhibitors of the signaling cascade.

## 1. Introduction

Glioblastoma, IDH wildtype, (GB) is a grade IV diffuse glioma [1], and represents the most common and aggressive primary brain tumor class in adults, with a median survival of 15 months and a 5-year survival rate of less than 5% [2]. Conventional treatment consists of surgical bulk removal of the tumor, followed up by combined radiotherapy and temozolomide (TMZ)-based chemotherapy [3]. However, current therapy protocols have low success, and patients almost always relapse, often in a more aggressive manner. This is partially due to a cellular hierarchy, in which a subpopulation of glioma stem cells (GSCs) contribute to tumor relapse [4] and therapeutic resistance [5,6]. Therefore, strategies aiming at the eradication of GSCs are potentially promising for improving the prognosis of patients with GB [7].

GSCs share properties with neural stem cells (NSCs). Studies in mouse models and human tumor specimens have demonstrated that NSCs are likely the cells of origin of human GB [8,9,10]. There is increasing evidence indicating that transcriptional and epigenetic pathways controlling normal NSCs activity also contribute to the regulation of GSCs [6,11]. Transcription factors that govern stem cell differentiation can potentially function as oncogenes by promoting the acquisition of transcriptional programs required for tumorigenesis, including epigenetic dysregulation [12]. Among them, several members of the SOX (Sex-determining region Y (SRY)-bOX) family of transcription factors have been identified to be essential for GB propagation and GSC activity [13,14,15,16].

SOX9 is a critical developmental regulator that plays an essential role in the establishment and maintenance of adult stem cells in a wide range of tissues, including the central nervous system [17]. Studies on *SOX9* gain and loss of function in cell cultures and mice models revealed that it maintains adult NSCs [18,19]. Moreover, SOX9 facilitates the neoplastic transformation of different cell types including NSCs [20,21,22] and exerts a pro-oncogenic activity by controlling cancer stem cells (CSCs) in several tumor types, including GB [14,23]. In tumor biopsies, SOX9 expression is generally elevated and correlates with poor prognosis [20,24]. Although it is known that the SOX9 transcription factor plays a role in CSCs, the SOX9-related molecular downstream effectors in these cells in human samples remain poorly understood.

In this study, we present evidence that SOX9 is a critical regulator for GSC maintenance. We reveal that STAT3 and PML are critical effectors regulated by SOX9. We found a genetic regulatory loop involving SOX9, STAT3, and PML that modulates GSC activity. Hence, pharmacological inhibition of the SOX9–STAT3–PML pathway may represent a novel treatment strategy for GB.

## 2. Results

### 2.1. High SOX9 Levels Correlate with Lower Patient Survival

We first characterized the expression of SOX9 in glioma grade taking advantage of publicly available TCGA and Rembrandt cohorts. Herein, we found that *SOX9* was higher in grade IV cases, linking its high levels to tumor malignancy (Figure 1A). Then, we analyzed the clinical relevance of SOX9 in GB by studying its expression in a cohort of 88 human GB patients from Donostia University Hospital and compared it with healthy brain tissue (Figure 1B). The levels of *SOX9* mRNA were significantly upregulated in GB, where more than 80% of samples (71 out of 88 patients) showed overexpression (fold change higher than two) (Figure 1C). These results were also confirmed at the protein level since TMA data from Donostia Hospital and University Hospital Basel showed that SOX9 expression was elevated in GB samples (Figure 1D,E). Remarkably, survival analysis revealed that GB patients with high SOX9 displayed significantly poorer outcomes than patients with low expression. Thus, patient median survival was reduced from 24 to 13 months in patients with high SOX9 from Donostia Hospital (Figure 1F) and from 8 to 3.5 months in patients from Hospital Basel (Figure 1G). These results suggest SOX9 as a prognostic biomarker in GB.

### 2.2. SOX9 Upregulation Increases Tumorigenic Capacity of GSCs

We previously demonstrated that SOX9 was overexpressed in oncospheres derived from conventional and patient-derived GSCs, both in vitro and in vivo [23]. In order to further ascertain the molecular and biological processes controlled by SOX9 in GSCs, we first overexpressed SOX9 in GNS166 patient-derived cells and also in U373-MG cells (Figure 2A) and performed in vitro and in vivo experiments. In vitro, SOX9 overexpression promoted increased cell growth (Figure 2B). In vivo, we observed an enhanced tumor formation capacity when SOX9-upregulated GNS166 cells were injected orthotopically in immunodeficient NOD-SCID mice. This led to a significant decrease in mice survival (*p* = 0.004), with an overall lifespan of 35 ± 7.8 days, compared with 116 ± 11.5 days of median survival in controls (Figure 2C). In addition, we injected control and SOX9-upregulated U373-MG cells subcutaneously in *Foxn1^nu^/Foxn1^nu^* nude mice, which notably increased their tumorigenic potential. Thus, 87.5% of mice inoculated with SOX9-upregulated U373-MG cells developed tumors, compared with the 12.5% of mice injected with control cells. Tumors generated by U373-MG cells with overexpression of SOX9 appeared earlier (day 53 vs. day 84), and they had greater volume (Figure 2D,E). Immunohistochemistry analyses also showed that these tumors had higher SOX9 and increased proliferative capacity measured by Ki67 expression (Figure 2F). Additionally, SOX2 levels were also increased suggesting enrichment of stemness activity (Figure 2F). Taken together, our results show that high SOX9 levels endorse tumorigenic capacity of GSCs.

### 2.3. SOX9 Downregulation Decreases Stemness and Tumorigenic Capacity in GSCs

To further address the impact of SOX9 on the regulation of GSCs, we knocked down *SOX9* in GNS166 cells and conventional cells U373-MG and U251-MG (with intermediate and high basal levels of SOX9, respectively). Data from immunoblotting experiments confirmed an effective reduction in SOX9 levels accompanied by a marked diminishment of SOX2 expression (Figure 3A). In this context, we detected a significant decrease of more than 50% in cell growth in U373-MG and U251-MG cells (Figure 3B), as well as a reduction in the number of phospho-Histone H3 (P-H3)-positive cells in GNS166 cells (Figure 3C) in SOX9 knockdown cells. This impairment in cell proliferation was accompanied by a significant increase in senescence measured by cytoplasmic SA-β-gal activity (Figure 3D) and also by a decrease in apoptosis [25]. Moreover, *SOX9* knockdown cells also presented markedly reduced stemness activity measured by decreased colony formation ability (Figure 3E), lower number of soft agar foci (Appendix A), increased expression of differentiation markers (Figure 3F), and reduced tumorsphere formation by more than 60% in both primary and secondary culture condition (Figure 3G and Appendix A).

We performed in vivo experiments and found a notably reduced tumor formation and progression capacity for *shSOX9* U373-MG cells in *Foxn1^nu^/Foxn1^nu^* nude mice. Thus, only 33% of mice inoculated with *shSOX9* cells developed tumors, compared with 85% of controls, and they formed smaller tumors (Figure 3H). In line with this, *shSOX9* tumors expressed lower levels of SOX9 and Ki67 than control tumors (Figure 3I). Similarly, SOX2 expression was also reduced (Figure 3I). Additionally, orthotopic injection of *shSOX9* GNS166 cells revealed a significant increase in mice survival (*p* = 0.009) passing from ~100 days in controls to almost 200 days in *shSOX9* (Figure 3J). Taken together, our results point out that SOX9 is a relevant mediator of malignant phenotypes of GB by modulating GSCs capacities.

### 2.4. Transcriptomic Analysis Reveals STAT3 and PML as Mediators of SOX9 Activity in GSCs

We addressed the molecular mechanisms underlying SOX9 function in GB. For that purpose, we compared expression microarray data from SOX9 knockdown GNS166 cells with control GNS166 cells. Cluster analysis showed differences in gene expression between both phenotypes being the levels of SOX9 gene the most decreased validating the experimental approach (Figure 4A and Appendix A). Gene Ontology analysis revealed JAK2-mediated signaling among the top pathways significantly altered in response to *SOX9* knockdown, together with interferon signaling, interleukin 17 (IL17) signaling, or growth hormones (Figure 4B). Janus kinase 2 (JAK2) is a tyrosine kinase that activates the signal transducer and activator of transcription 3 (STAT3) transcription factor, which has been involved in the progression of most types of cancers including GB, which is required for GSC growth and self-renewal [26]. In addition, activated STAT3 levels are known to correlate with promyelocytic leukemia (*PML*) gene expression in several tumor models [27], including breast cancer, where it acts as an upstream regulator of *PML* [28]. In this context, we hypothesized that STAT3 and PML could be part of a regulatory pathway that regulates GSC activity. We validated transcriptomic results and detected reduced levels of STAT3 (total STAT3 and p-STAT3), as well as PML in *shSOX9* GNS166 cells (Figure 4C). Similar results were obtained in *SOX9*-silenced U251-MG cells (Figure 4C). Accordingly, moderately higher levels of STAT3, p-STAT3, and PML were detected in SOX9 overexpressing U373-MG cells (Figure 4D). These data suggest that SOX9 regulates STAT3 and PML expression in GB.

We analyzed SOX9, STAT3/p-STAT3, and PML expression in a set of established glioma cell lines (Figure 4E) and patient-derived cells cultures (Figure 4F,G) by immunoblot. We observed a positive correlation among SOX9, p-STAT3, and PML protein levels in all cell lines except U373-MG cells (Figure 4E–G). This correlation was also identified in tumorspheres from U87-MG and U251-MG cells, where SOX9, p-STAT3, and PML expression was increased (Figure 4H). In addition, similar results were obtained in subcutaneous tumors generated from U373-MG secondary tumorspheres in which the number of positive cells and the intensity of staining is higher for SOX9, STAT3, and PML, compared with U373-MG parental cells (Figure 4I) [23]. In line with this, the expression of STAT3 and PML also was higher in grade IV clinical biopsies from publicly available TCGA and Rembrandt cohorts (Figure 4J).

To further reinforce the link between SOX9, STAT3, and PML, we examined bioptic tissue. Remarkably, the association was further confirmed in GB samples from the Hospital Basel cohort, where we found a significant positive correlation between SOX9 with PML and p-STAT3, as well as between PML and p-STAT3 (Figure 4K). To reinforce this result, we compared *SOX9* and *STAT3*, as well as *PML* and *STAT3* mRNA expression in The Cancer Genome Atlas (TCGA) cohort (*n* = 580) [29] and again found a positive correlation (ρ Spearman = 0.61 and 0.68, respectively) (Figure 4L). Together, these results reveal that SOX9, STAT3, and PML expression correlate in GB biopsies.

### 2.5. STAT3 Regulates GSC Activity and Its Pharmacological Inhibition Reduces Tumorigenicity

To investigate the SOX9–STAT3–PML axis as a potential therapeutic target, we knocked down *STAT3* in GNS166 and U251-MG cells using two different *shRNA* constructs (*sh41* and *sh43*) and analyzed the resulting phenotypes. Immunoblot confirmed the reduced levels of STAT3 and p-STAT3, as well as SOX9 and PML (Figure 5A). *STAT3* silencing led to a significant reduction of more than 50% in cell proliferation in both models (Figure 5B), which was associated with an increased number of senescent cells (Figure 5C). Moreover, *STAT3* knockdown impaired the tumorsphere formation of U251-MG glioma cells (Figure 5D). These results confirm overlapping phenotypes in *SOX9* and *STAT3* knockdown glioma cells in vitro.

We next moved to in vivo assays, which further confirmed these results. Thus, immunodeficient mice subcutaneously injected with *sh41* or *sh43* U251-MG cells did not form tumors (*sh43*) or formed them at a low percentage and with less volume (*sh41*) than control mice (Figure 5E). In addition, the limiting dilution assay revealed that *STAT3* knockdown led to a reduced number of tumor-initiating cells (Figure 5F). Overall, our data reveal that *STAT3* knockdown produces equal phenotypes of similar severity as those observed after *SOX9* downregulation.

Since established pharmacological STAT3 inhibitors could potentially be translated to clinical practice, we explored the effect of STAT3 inhibition on the SOX9–STAT3–PML axis and GB progression. We tested the specific STAT3 inhibitor *STX-0119*, which prevents the dimerization of STAT3 and its subsequent binding to DNA. We found that increasing doses (25, 50, and 100 µM) of this drug significantly reduced viability in glioma and GSC cells (Figure 5G). In addition, the tumorsphere formation ability rate was significantly diminished in all cell cultures (Figure 5D). Remarkably, the pharmacological inhibition of STAT3 reduced SOX9 and PML expression levels, as well as the expression of SOX2, a stem cell marker (Figure 5I), mimicking the effects of gene silencing of *STAT3*. We have previously described that SOX2 controls SOX9 levels [23], so it might be of interest to unravel whether it plays a role in the SOX9–STAT3–PML axis.

### 2.6. PML Regulates GSC Activity and Its Pharmacological Inhibition Reduces Tumorigenicity

We characterized the role of PML in GBM and knocked down *PML* expression in GNS166 and U251-MG cells, using two different *shRNA* (*shPML-1* and *shPML-4*). Immunoblot confirmed the effective silencing of *PML* (Figure 6A). At a functional level, PML knockdown promoted a significant decrease in cell growth, ranging from 30% to 80%, depending on the *shRNA* and cell type (Figure 6B). This reduction was also observed when analyzing the number of P-H3-positive cells in both cases (Figure 6C), which was accompanied by a significant increase in cellular senescence (Figure 6D). In addition, *PML* knockdown impaired tumorsphere formation and self-renewal activity in U251-MG cells (Figure 6E). These results confirm overlapping phenotypes in *SOX9*, *STAT3*, and *PML* knockdown glioma cells in vitro.

Further, *PML* knockdown in GNS166 cells reduced their tumorigenic capacity when they were injected orthotopically in immunodeficient mice. Thus, *shPML-4* cells showed a significant increase in mice survival (*p* = 0.005), with an overall lifespan of 116 ± 11.5 days, compared with 174.5 ± 36.9 days median survival of controls (Figure 6F). These data confirm that PML is essential to sustain GSC activity and that *PML* silencing recapitulates the effects observed with SOX9 and STAT3 genetic inhibition.

Arsenic trioxide (ATO) is a pharmacological compound that induces SUMO-dependent ubiquitylation and proteasome-mediated degradation of the PML protein [30]. We explored the impact of pharmacological inhibition of PML using ATO in glioma cells. First, we treated U87-MG and U251-MG glioma cell lines with increasing doses (0.5, 1, and 5 µM) and observed that ATO promoted a significant cytotoxic effect in both cell lines in a dose-dependent manner (Figure 6G). Further, we analyzed the effect of ATO in the formation of tumorspheres and detected a reduction, particularly for U87-MG cells (Figure 6H). Notably, the expression of PML, together with SOX9, was reduced at the protein level in a dose-dependent manner (Figure 6I). These data show that both genetic and pharmacological inhibition of PML have effects on glioma cell activity, comparable to those observed when *SOX9* and *STAT3* were silenced.

### 2.7. The SOX9–STAT3–PML Axis Constitutes a Regulatory Loop That Modulates GSCs

Since the expression of SOX9 was altered by genetic and pharmacological inhibition of STAT3 and PML, we hypothesized that SOX9–STAT3–PML could represent a signaling axis to regulate GSC activity in GB. To evaluate this, we simultaneously silenced *SOX9* and *PML* in GNS166 cells (using *shSOX9-1* and *shPML-4* plasmids). Quantification at the mRNA level confirmed the successful inhibition of both genes with no additive effects (Figure 7A,B). This co-inhibition of *SOX9* and *PML* led to a decrease in cell proliferation but not significantly higher than individual knockdown (Figure 7C). Similar effects on senescence were observed after individual silencing, compared with dual inhibition of *SOX9* and *PML* (Figure 7D). These data support the idea that SOX9 and PML act in the same pathway and that a regulatory loop between both genes might exist.

To further confirm this idea, we ectopically overexpressed *SOX9* in GNS166 cells previously silenced for *PML* and analyzed the resulting phenotypes. Immunoblot in GNS166 cells showed that SOX9 restoration in the absence of PML was able to re-establish the expression of SOX9 and PML (Figure 7F), suggesting that a feedback loop may exist among both genes. We further investigated whether SOX9 reactivation restored the phenotype associated with PML knockdown and found that cell proliferation was significantly increased in the presence of ectopic SOX9 in vitro (Figure 7G). In addition, SOX9 overexpression in *PML*-silenced GNS166 cells enhanced their tumorigenic activity reaching levels of control cells, since immunodeficient mice orthotopically injected with *shPML4/SOX9* cells showed an overall lifespan of 133 days, compared with 174.5 days of *shPML4* cells (Figure 7H).

To determine the mechanism by which PML regulates SOX9 activity, we explored a potential transcriptional regulation. For that, we performed a ChIP analysis of ectopically expressed (doxycycline-inducible HA-PMLIV) and endogenous *PML* in U251-MG cells. As a result, *SOX9* promoter regulatory sequences were found and PML bound to the SOX9 promoter (Figure 7E), confirming that PML directly modulates *SOX9* transcription in GB cells. Taken together, our results suggest that SOX9, STAT3, and PML participate in a common regulatory loop that contributes to maintaining GSC activity.

## 3. Discussion

GB is a currently incurable tumor with a poor prognosis. Its diffuse brain infiltration, which precludes total resections and the existence of GSCs, contributes to recurrence and therapy resistance.

SOX9 is a transcription factor and a pioneer factor critically involved in stem cell maintenance and plasticity [31]. We had previously reported that SOX9 is highly expressed in GSCs [23], which is supported by recent studies on GB [32]. Gain and loss of function of SOX9 demonstrate a link between SOX9 and GSCs, where this transcription factor is essential for maintaining stemness properties. We observed a clinical correlation between high SOX9 expression levels and lower survival in patient biopsies from two independent GB cohorts, in agreement with existing data [24,33]. Our results provide novel evidence concerning the important role of SOX9 in GSC activity and GB progression, prompting for deciphering the SOX9-related molecular network in GSCs and GB.

For the first time, we performed a transcriptomic analysis of SOX9 targets in patient-derived GSCs identifying several candidates that specifically impact intrinsic properties of GSCs such as self-renewing and tumorigenic activity. In particular, we identified STAT3, which is a transcription factor with a central role in many types of cancer including GB, where it represents a convergence point of oncogenic pathways [26,34]. We observed that STAT3 is regulated by SOX9 in GSCs, and their expression is correlated in GB biopsies. Reinforcing our results, higher levels of STAT3 in GB biopsies, compared with healthy brain tissue, have also been reported [34], and are associated with poorer prognosis [35,36]. Interestingly, *STAT3* knockdown in GSCs and conventional glioma cell lines coincided with the phenotypes described with *SOX9* silencing of reduced self-renewing in vitro and decreased tumorigenic activity in vivo. In line with our results, STAT3 promotes GSC maintenance, tumor invasion, or immune evasion in GB [35].

We have previously seen that STAT3 regulates PML and SOX9 in metastatic breast cancer [28]. PML is a key regulator of CSCs maintenance that promotes tumorigenesis [37,38]. Our results point out that *PML* acts as an oncogene, which is required to maintain GSC activity, together with SOX9 and STAT3. In line with our results, a previous study showed that *PML* exerts an oncogenic role in GB [39]. Notably, our results show, for the first time, that a regulatory loop, SOX9–STAT3–PML axis, modulates GSCs. In our previous research, we identified that PML directly regulates *SOX9* transcription [28], and STAT3 is upstream of PML. However, no link was observed between SOX9 and STAT3. Therefore, the SOX9–STAT3–PML axis may be critical for tumorigenesis. This might open new avenues in the treatment of a variety of tumors by targeting a common oncogenic pathway.

Our finding that SOX9 activity is part of the same oncogenic signaling pathway as that for STAT3 and PML proposes that pharmacological inhibition of this axis could represent a strategy to target the GSC subpopulation. In the case of SOX9, we had previously demonstrated that rapamycin reduces SOX9 expression in GB, and a combined treatment of rapamycin with TMZ enhances this effect [23]. Regarding STAT3, we observed that STX-0119 inhibits the SOX9–STAT3–PML expression, and thereby GSC cell growth and self-renewing, in line with a previous study in stem-like cell lines derived from human GB [40]. Interestingly, in vivo experiments performed in MHC-double knockout mice xenografted with TMZ-resistant U87 cells that were further treated with STX-0119 also showed a relevant inhibitory effect of this drug on tumor growth [41]. Other drugs targeting JAK2/STAT3 signaling have been successfully tested in different GB models [42,43], which supports our results. A phase I clinical trial (NCT01904123) with the JAK2 inhibitor WP1066 in patients with recurrent malignant glioma is ongoing. In the case of PML, we report that ATO impacts PML and SOX9 expression and reduces cell proliferation and GSCs’ intrinsic properties of self-renewing and tumorigenic activity. Our functional results are in concordance with previous results, also in GSCs [44]. Remarkably, ATO is being tested in combination with TMZ and radiotherapy in phase I/II trial (NCT00275067) on patients with surgical resection of malignant glioma. The potential of PML pharmacological inhibition with ATO is reinforced by a recent study showing that PML expression is required to sensitize primary human GBM cells to ATO treatment [39].

GBs are divided into different subtypes based on molecular signatures [45] and are known to derive from different cell types including for neoplastic transformation of NSCs [8,10,46]. Notably, a recent study from Dr. Luis Parada’s Lab has proposed a new stratification for GB patients in type I (derived from NSCs) and type II subtypes based on different species-conserved transcriptional profiles with unique therapeutic sensitivities for type II tumors [46]. However, no specific drugs were reported to be effective for type I GB, which is characterized by high expression of SOX9 and EGFR, raising SOX9 as a key driver of this GB subset [46]. Moreover, high levels of STAT3 signaling have been linked to mesenchymal subtype and are responsible for immune evasion [47]. Even though current epigenomic glioma classification [48] was unavailable for our large cohort of retrospective specimens which in turn have adequately extensive clinical follow-up data associated, our results strongly support our pharmacological approach of inhibiting the SOX9–STAT3–PML axis as a new therapeutic strategy in GB, at least for a subset of patients originated from the transformation of NSCs, for mesenchymal subtype or with high levels of SOX9, STAT3 and/or PML. In this sense, reliable and accurate stratification criteria to classify patients for personalized treatment are required. Our cut-off point of 60% of SOX9-positive cells to stratify GB patients based on their SOX9 levels was significantly associated with survival in two independent cohorts. A STAT3-based gene signature also has been shown to stratify glioma patients for targeted therapy [35]. Given current molecular classification tools such as epigenomic profiling, it might be worthwhile to prospectively document the SOX9–STAT3–PML axis and correlate its activity levels with current high-grade glioma methylation classes.

Taken together, we provide strong evidence that SOX9 and its targets STAT3 and PML form a molecular axis, which is an important contributor to GB pathology through their role as regulators of GSCs. In addition, we propose SOX9–STAT3–PML as useful prognostic biomarkers of both GB and GSCs that represent promising therapeutic targets in a subset of patients showing high expression levels.

## 4. Materials and Methods

### 4.1. Human Subjects

This study included biopsies and clinical information from 88 patients examined at Donostia University Hospital (San Sebastian, Spain), histologically diagnosed as glioblastoma, IDH wildtype (WHO grade IV). Human GB samples were provided by the Basque Biobank for Research-OEHUN (http://www.biobancovasco.org/, accessed on 10 March 2022). In addition, an independent cohort of 20 GB patients from the University Hospital Basel (Switzerland) was analyzed. Transcriptomic data were collected from the GlioVis database (http://gliovis.bioinfo.cnio.es/, accessed on 9 December 2021). This study was approved by the Basque Clinical Research Ethics Committee (07/2013) and adhered to the tenets of the Declaration of Helsinki by the World Medical Association regarding human experimentation. All participants read and signed an informed consent form.

Given the acquisition timeline of the tumor specimens and associated clinical long-term follow-up that predate the 2016 WHO classification [1], the definition of glioblastoma, IDH wildtype is not as strict as currently defined in the upcoming 2021 WHO classification of central nervous system tumors.

### 4.2. Tissue Microarrays

Tissue microarrays (TMAs) were used to determine SOX9, phosphorylated STAT3 (p-STAT3), and PML expression in GB patients from two independent cohorts from Donostia University Hospital (*n* = 47) and from University Hospital Basel (*n* = 20). SOX9 expression was ranked as “SOX9^+^” (less than 60% of positive cells) and “SOX9^++^” (more than 60%). Quantification of p-STAT3 was based on “p-STAT3 0” (absence of staining), “p-STAT3 1” (≈1–30% positive cells), “p-STAT3 2” (≈30–60%), and “p-STAT3 3” (more than 60%). PML quantification was based on the number of stained nuclear points: “PML 0” (absence of staining), “PML 1” (≈1–10%), “PML 2” (≈10–30%), and “PML 3” (more than 30%).

### 4.3. Cell Culture

Glioma cell lines U87-MG, U251-MG, U373-MG, A172, and T98G were purchased from the ATCC (American Type Culture Collection). Patient-derived GNS166 and GNS179 cell lines were kindly provided by Steven Pollard´s Laboratory (University of Edinburgh) and grown as previously reported [49], and GB1 and GB2 were generated in the laboratory. All cell lines were mycoplasma-free and authenticated by *GenePrint10System Kit* (Promega, Madison, WI, USA). Glioma cell lines were cultured as adherent monolayers in Dulbecco’s modified Eagle medium (DMEM) media (cat#11995065, Life Technologies, Carlsbad, CA, USA) supplemented with 10% fetal bovine serum (FBS) (Life Technologies, Carlsbad, CA, USA), or as oncospheres in DMEM/F-12 medium (cat#10565018, Life Technologies, Carlsbad, CA, USA) supplemented with glucose 45% (Sigma-Aldrich, St. Louis, MO, USA), growth factors (20 ng/mL epidermal growth factor (EGF) and 20 ng/mL basic fibroblast growth factor (bFGF); Sigma-Aldrich), and N2 and B27 supplements (Fisher, Waltham, MA, USA). Patient-derived cell lines were also grown in this medium. Before seeding, culture plates for GNS cells were treated with 10 μg/mL of laminin (Sigma-Aldrich) for 3 h at 37  °C. Cells were maintained under standard conditions in a humidified atmosphere of 5% CO_2_ at 37  °C.

### 4.4. Gene Silencing and Overexpression

For *SOX9* silencing, cells were infected with lentiviral particles harboring the *pLKO.1-sh-hSOX9-1* plasmid (gift from Dr. Bob Weinberg; Addgene plasmid #40644). For *STAT3* downregulation, cells were transduced with lentivirus containing *sh41 STAT3* (Sigma-Aldrich TRCN0000020841) or *sh43 STAT3* (Sigma-Aldrich TRCN0000020843) plasmids. To inhibit *PML* expression, lentiviral particles harboring *shPML-1* (Sigma-Aldrich TRCN0000003865) or *shPML-4* (Sigma-Aldrich TRCN 0000003867) plasmids were used. The *pLKO.1 puro* empty vector (gift from Dr. Bob Weinberg; Addgene plasmid #8453) was used as a control in all cases. Transduced cells were selected in the presence of 2 μg/mL puromycin (Sigma-Aldrich) for 48 h and then maintained with 0.2 μg/mL puromycin. For lentiviral *SOX9* overexpression, the *pWPXL-SOX9* plasmid (gift from Dr. Bob Weinberg; Addgene plasmid #36979) and a control plasmid harboring GFP (*pWXL GFP*, a gift from Dr. Didier Trono; Addgene plasmid #12257) were used. Lentiviral infections were performed following standard procedures at a multiplicity of infection (MOI) of 10 for 6 h.

### 4.5. Cell Proliferation Assay

Cell growth was evaluated following standard procedures. Briefly, 2.5 × 10^4^ cells per well in 6-well plates were seeded in duplicate, and total cell count was performed on days 1, 3, and 5.

### 4.6. Oncosphere Formation Assay

In total, 500 cells per well were seeded in untreated, 12-well, flat-bottom plates containing oncosphere medium; fresh medium was added every three days. After 10 days in culture, spheres were counted using a light microscope. For secondary (*2^ry^*) generation, spheres were mechanically and enzymatically disaggregated with accutase (Life Technologies, Carlsbad, CA, USA) and then maintained in culture for another 10 days.

### 4.7. Senescence-Associated β-Galactosidase Staining

To evaluate cellular senescence, the senescence-associated β-galactosidase (SA-β-gal) activity was measured using a commercial staining kit (9860S, Cell Signaling, Danvers, MA, USA) according to the manufacturer’s protocol.

### 4.8. Cell Viability Assay

Cells were seeded in 96-well plates at a density of 2.5 × 10^3^ cells per well and treated 24 h later with the indicated concentrations of STX-0119 (Sigma-Aldrich) for 72 h or arsenic trioxide (ATO; Sigma-Aldrich) for 48 h, in sextuplicates. As a control, cells were treated with the indicated solvent for each drug. Cells were then incubated with 0.25 mg/mL 3-(4,5-dimethylthiazol-2-yl)-2,5-diphenyltetrazolium bromide (MTT; Sigma-Aldrich) for 3 h. The formazan produced by viable cells was dissolved in 150 μL of dimethyl sulfoxide (DMSO; Sigma-Aldrich), and the absorbance was determined at 570 nm in a microplate reader (Multiskan Ascent Thermo Electron Corporation, Waltham, MA, USA).

### 4.9. Quantitative Real-Time PCR

Total RNA was extracted using Trizol (Life Technologies). Reverse transcription was performed from 1 μg of total RNA using random primers and the High-Capacity cDNA Reverse Transcription Kit (Life Technologies) according to the manufacturer’s guidelines. Quantitative real-time PCR (qPCR) was then carried out using Power SYBR^®^ Green Master Mix (Thermo Scientific, Waltham, MA, USA), 10 mM of primers, and 20 ng of cDNA in an ABI PRISM 7300 thermocycler (Applied Biosystems, Waltham, MA, USA). The ΔΔCT method was used for relative quantification of gene expression, using *GAPDH* as the reference gene. Primer sequences are available upon request.

### 4.10. Immunoblot

Immunoblots were performed as previously described [50]. The following primary antibodies were used: rabbit polyclonal anti-SOX9 (1:2000 dilution; Millipore, Burlington, MA, USA), mouse monoclonal anti-STAT3 (1:1000 dilution; Cell Signaling), rabbit monoclonal anti-phospho STAT3 (Tyr705) (1:1000 dilution; Cell Signaling), rabbit polyclonal anti-PML (1:1000 dilution; Bethyl laboratories, Montgomery, TX, USA), rabbit polyclonal anti-SOX2 (1:500 dilution; Millipore), and mouse monoclonal anti-β actin (1:2000 dilution; Sigma-Aldrich). Primary antibodies were detected with horseradish peroxidase (HRP)-linked antibodies: Goat anti-rabbit or goat anti-mouse (Santa Cruz Biotechnology, Dallas, TX, USA). Protein detection was performed using the NOVEX^®^ ECL system (Invitrogen, Carlsbad, CA, USA).

### 4.11. Immunofluorescence

A total of 2 × 10^4^ cells were seeded in Lab-Tek II Chamber Slides (ThermoFisher), fixed after 24 h with 4% formalin for 10 min at room temperature (RT), and blocked with phosphate-buffered saline (PBS) supplemented with 0.3% Triton X-100 (Sigma-Aldrich) and 5% FBS for 1 h at RT. Cells were then incubated for 2 h at RT with mouse monoclonal anti-Histone H3 (1:1000 dilution; Abcam, Cambridge, UK) and with Alexa Fluor^®^ 488 rabbit anti-mouse IgG (H + L) secondary antibody (1:500 dilution; A-11059, Invitrogen) for 1 h at RT. After that, slides were washed and mounted with a Vectashield Mounting Medium with DAPI (Vector Laboratories, Burlingame, CA, USA). Images were obtained with a Nikon Eclipse 80i microscope.

### 4.12. In Vivo Carcinogenesis Assays

For subcutaneous injection, cells were resuspended in PBS. Briefly, 1 × 10^6^ cells per condition were injected subcutaneously into both flanks of 8-week-old *Foxn1^nu^/Foxn1^nu^* nude mice. External calipers were used to measure tumor size; tumor volume was then estimated by the following formula: V = L × W^2^ × 0.5 (L = tumor length and W = tumor width). For xenotransplantation, GSCs were injected stereotactically into the frontal cortex of 6–8-week-old NOD-SCID mice. Briefly, GSCs were disaggregated with accutase and resuspended in PBS. Then, 1 × 10^4^ or 1 × 10^5^ cells in a final volume of 1 µL were injected into the putamen using a stereotaxic apparatus (Kopf Instruments). For tumor initiation assays, U251-MG cells at a density of 5 × 10^4^ or 5 × 10^5^ cells per condition were injected subcutaneously into both flanks of 8-week-old *Foxn1^nu^/Foxn1^nu^* nude mice. To calculate the number of initiating cells, we used Extreme Limiting Dilution Analysis (ELDA) software (http://bioinf.wehi.edu.au/software/elda/, accessed on 10 March 2022). 

All animal handling and procedures were performed according to the ethical guidelines established by the Animal Experimentation Ethics Committee of Biodonostia Health Research Institute (CEEA14/016) and conducted in conformity with the European Union recommendations for animal experimentation specified in the Directive 2010/63/EU.

### 4.13. Immunohistochemistry

Tumors generated in mice were dissected, fixed in 10% formalin for 48 h, and embedded in paraffin. Four micrometer-thick sections were stained with hematoxylin–eosin (H&E) using a Varistain Gemini ES machine (ThermoFisher, Waltham, MA, USA). For immunohistochemistry (IHC), sections were rehydrated and heated in citrate buffer pH 6 for 10 min for antigen retrieval. Endogenous peroxidase was blocked with 5% hydrogen peroxide in methanol for 15 min. Sections were incubated with the following primary antibodies: rabbit polyclonal anti-SOX9 (1:1000 dilution; Millipore), mouse monoclonal anti-phospho-STAT3 (Tyr705) (1:100 dilution; Cell Signaling), mouse monoclonal anti-PML (PG-M3) (1:200 dilution; Santa Cruz Biotechnology), rabbit polyclonal anti-SOX2 (1:500 dilution; Millipore), and rabbit polyclonal anti-Ki67 (1:1000 dilution; Abcam). Sections were then incubated with MACH 3 Rabbit HRP-Polymer (BioCare Medical). Staining was developed with 3,3’-diaminobenzidine (DAB) (Spring Bioscience, Pleasanton, CA, USA). IHC images were obtained with a Nikon Eclipse 80i microscope.

### 4.14. Transcriptome Analysis

Expression microarrays were performed from 0.5 μg of RNA from *SOX9*-silenced GNS166 cells and control GNS166 cells using the *Gene Chip Human Genome U133 Plus 2.0* array (Affymetrix). Data were normalized by Robust Multiarray Average (RMA) using *Affymetrix^®^ Expression Console™* software V1.1. Differential gene expression analysis was carried out with *Genespring* software GeneSpring GX 14.9 (Agilent, Santa Clara, CA, USA). Pathway analysis was performed with the *Interactive Pathway Analysis* (IPA^®^) software (Ingenuity Systems, Redwood city, CA, USA, https://www.qiagenbioinformatics.com/products/ingenuity-pathway-analysis, accessed on 10 March 2022). The data discussed in this publication have been deposited in NCBI’s Gene Expression Omnibus and are accessible in GSE181035.

### 4.15. Chromatin Immunoprecipitation Assay

A *SimpleChIP Enzymatic Chromatin IP Kit* (Cell Signaling) was used for the chromatin immunoprecipitation (ChIP) assay. In detail, U251-MG cells harboring a doxycycline-inducible HA-PMLIV plasmid were grown in 150 mm dishes for 3 days and cross-linked with 35% formaldehyde for 10 min at RT. Then, cells were incubated for 5 min at RT after the addition of glycine, washed with PBS, and scraped into PBS with 100 µM phenylmethylsulfonyl fluoride (PMSF). Pelleted cells were lysed and nuclei harvested. Nuclear lysates were digested with micrococcal nuclease for 20 min at 37 °C and then sonicated in 500 μL aliquots on ice for 3 pulses of 15 s using a Branson sonicator. Lysates were clarified, and chromatin was stored at −80 °C until used. HA-Tag polyclonal antibody (Cell Signaling), rabbit polyclonal anti-PML (Bethyl laboratories), and normal rabbit IgG polyclonal antibody (Cell Signaling) were incubated overnight at 4 °C with rotation. After that, protein G-conjugated magnetic beads were added and incubated for 2 h at 4 °C. Washes and elution of chromatin were later performed and DNA quantification was carried out using a Viia7 Real-Time PCR System (Applied Biosystems) with SYBR Green and primers that amplify the predicted PML binding region to *SOX9* promoter (chr17:70117013-70117409). Primer sequences used were forward 5′-ccggaaacttttctttgcag-3′ and reverse 5′-cggcgagcacttaggaag-3′.

### 4.16. Data Availability Statement

The authors confirm that the data supporting the findings of this study are available within the article and its Appendix A. The microarray data that support the findings of this study are openly available in NCBI’s Gene Expression Omnibus repository.

### 4.17. Data Analysis

Data are presented as the mean ± standard error of the mean (SEM) with the number of experiments (n) in parentheses. Mean values from quantitative variables were compared using Student’s *t*-test. Log-rank test was performed for Kaplan–Meier survival analyses. Correlations were calculated using the Spearman coefficient. Chi-squared (X^2^) test was used to compare frequency data. Asterisks (*, ** and ***) indicate statistical significance (*p* < 0.05, *p* < 0.01 and *p* < 0.001, respectively).

## Figures and Tables

**Figure 1 ijms-23-04511-f001:**
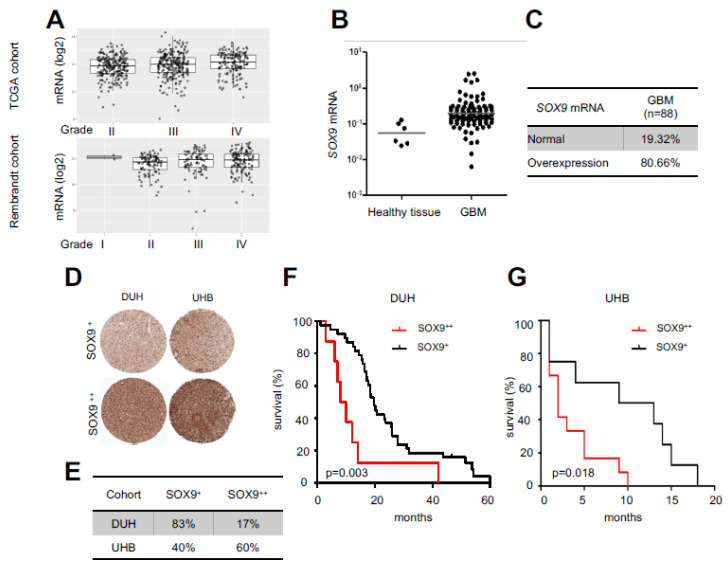
High levels of SOX9 correlate with poor survival: (**A**) *SOX9* mRNA expression in grade II, III, and IV of glioma in TCGA and Rembrandt cohorts; (**B**) *SOX9* mRNA expression in GB patients from the Donostia University Hospital (DUH; *n* = 88) relative to the mean expression in healthy brain tissue (*n* = 6); (**C**) percentage of patients with “overexpression” and “normal” expression of *SOX9* in the DUH cohort; (**D**) representative images of TMAs for SOX9 staining in GB samples from the DUH (*n* = 47) and University Hospital Basel (UHB; *n* = 20) cohorts; (**E**) percentage of patients with “SOX9^+^” (less than 60% of SOX9-positive cells) and “SOX9^++^” (equal to or more than 60%) expression from TMAs shown in “C”; (**F**) Kaplan–Meier survival analysis in GB patients from the DUH cohort (*n* = 47; *p* = 0.003) and (**G**) UHB cohort (*n* = 20; *p* = 0.018) based on SOX9 protein expression levels determined by TMA.

**Figure 2 ijms-23-04511-f002:**
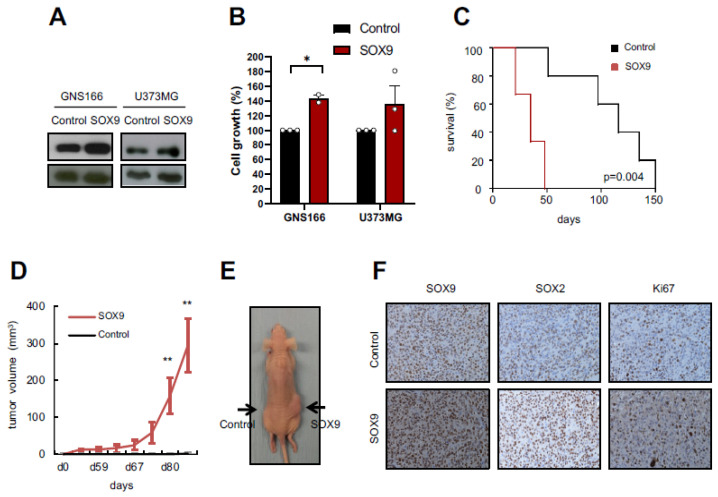
SOX9 upregulation leads to increased tumorigenicity: (**A**) representative immunoblot of SOX9 expression in indicated glioma cells infected with *pWXL GFP* (“Control”) and *pWPXL-Sox9* (“SOX9”) plasmids. β-actin was used as loading control; (**B**) quantification of cell growth in control and *SOX9* overexpression in U373-MG and GNS166 cells at day 5; (**C**) Kaplan–Meier curves representing NOD-SCID mice survival after stereotactic injection of GNS166 control and SOX9 GNS166 cells (*n* = 4); (**D**) tumor volume at indicated time points after subcutaneous injection of control and SOX9 U373-MG cells in immunodeficient mice (*n* = 4); (**E**) representative image of tumors in both flanks (right flank: injection of SOX9-overexpressed cells; left flank: control cells). Tumors are indicated by arrows; (**F**) representative images of SOX9, SOX2, and Ki67 IHC staining from subcutaneously generated tumors in “Control” and “SOX9” groups. Scale bar corresponds to 100 µm. * *p* < 0.05, ** *p* < 0.01.

**Figure 3 ijms-23-04511-f003:**
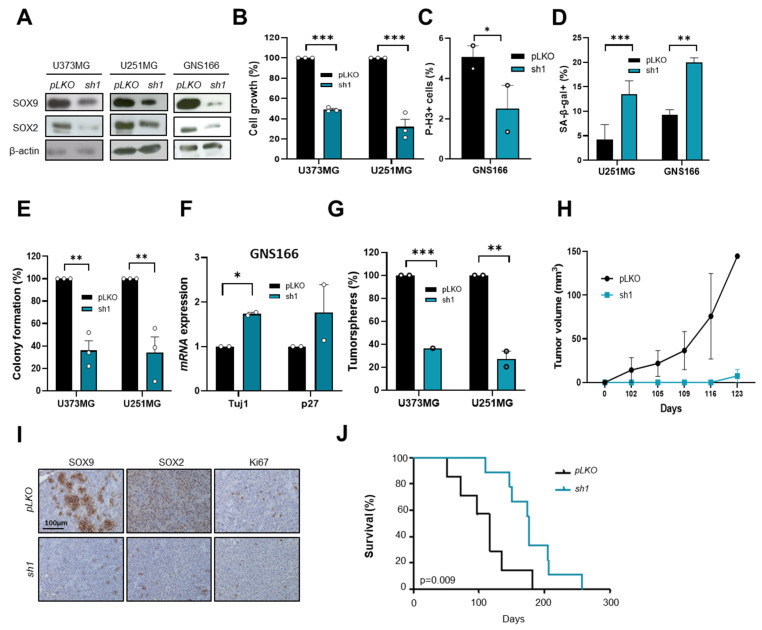
SOX9 knockdown impairs tumorigenicity: (**A**) representative immunoblot of SOX9 and SOX2 expression in glioma cells with SOX9 downregulation (*shSOX9-1*; *sh1*) and controls (*pLKO*). β-actin was used as loading control (*n* = 3); (**B**) quantification of cell growth in control and *sh1* U373-MG, U251-MG cells at day 5 (*n* = 3); (**C**) quantification of P-H3^+^ GNS166 cells infected with *pLKO* or *sh1* plasmids (*n* = 3); (**D**) quantification of SA-β-gal^+^ cells in *sh1* and control cells (*n* = 3); (**E**) quantification of number of colonies formed from glioma cell lines infected with *pLKO* or *sh1* plasmids (*n* = 3); (**F**) expression of *Tuj1* and *p27^KIP^* in GNS166 cells infected with *pLKO* or *sh1* plasmids (*n* = 2); (**G**) quantification of primary tumorspheres after 10 days in culture (*n* = 3). Quantifications are expressed relative to *pLKO*; (**H**) tumor volume of subcutaneous tumors generated by *pLKO*- and *sh1*-infected U373-MG cells (*n* = 4); (**I**) SOX9, SOX2, and Ki67 IHC staining from subcutaneously generated tumors in “*pLKO*” and “*sh1*” groups. Scale bar corresponds to 100 µm; (**J**) Kaplan–Meier curves representing mice survival after stereotactic injection of *pLKO* and *sh1* GNS166 cells. * *p* < 0.05, ** *p* < 0.01, *** *p* < 0.001.

**Figure 4 ijms-23-04511-f004:**
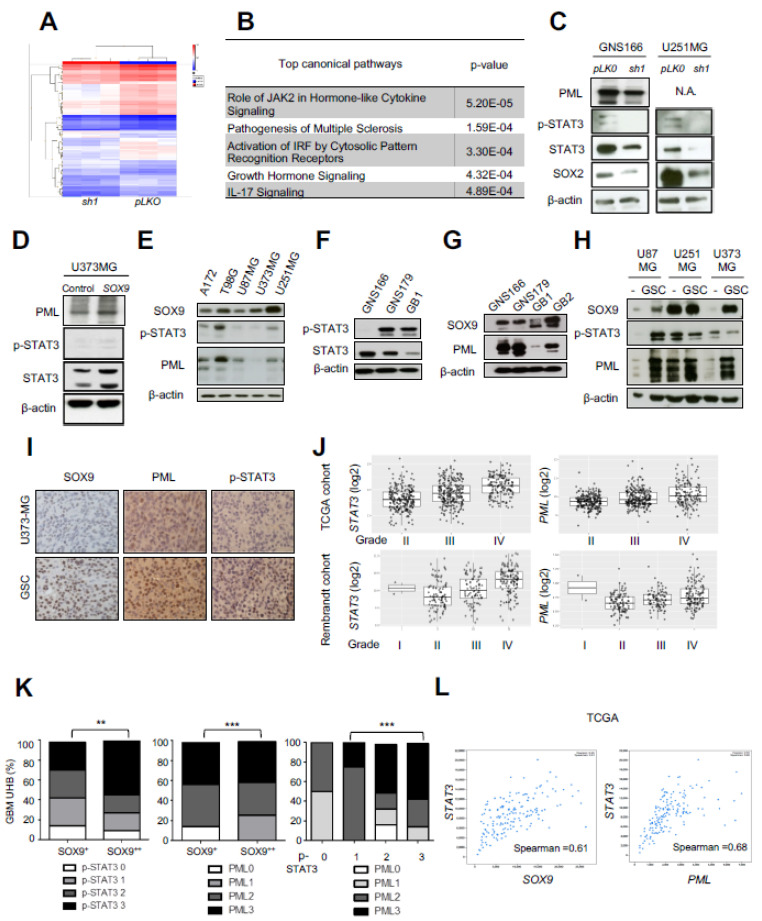
SOX9 expression correlates with STAT3 and PML: (**A**) cluster analysis of GNS166 cells after transcriptomic analysis; (**B**) top canonical pathways altered in *shSOX9* (*sh1*) GNS166 cells compared to controls (*n* = 3); (**C**) representative immunoblots of PML, p-STAT3, STAT3 and SOX2 expression in *sh1* and *pLKO* GNS166 and U251-MG cells (*n* = 3); (**D**) representative immunoblot of PML, p-STAT3, and STAT3 expression in SOX9 and control U373-MG cells (*n* = 3); (**E**) representative immunoblot of SOX9, p-STAT3, and PML expression in indicated glioma cell lines (*n* = 3); (**F**) representative immunoblot of p-STAT3 and STAT3 expression in GNS166, GNS179, and GB1 patient-derived glioma stem cells; (**G**) representative immunoblot of SOX9 and PML protein levels in patient-derived GSCs (23) (*n* = 3); (**H**) representative immunoblot of SOX9, p-STAT3, and PML expression in parental (“-”) and oncospheres (GSCs) from indicated cells; (**I**) representative images of SOX9, PML, and p-STAT3 IHC staining from tumors after subcutaneous injection of parental and tumorspheres from U373-MG cells. Scale bar corresponds to 50 µm; (**J**) *STAT3* and *PML* mRNA expression in grades II, III, and IV of glioma in TCGA and Rembrandt cohorts; (**K**) TMA of p-STAT3, PML, and SOX9 in GB human samples from the Hospital Basel cohort (*n* = 20). Chi-squared (χ^2^) test was used to compare frequency data and showed statistically significant differences between groups (*p* < 0.01 or 0.0001, respectively) with high and low staining of the indicated proteins; (**L**) correlation of *STAT3* with *SOX9* and *PML* mRNA expression in GB samples from the TCGA cohort (*n* = 580) (ρ Spearman = 0.61 and 0.68, respectively). N.A., not available. ** *p* < 0.01, *** *p* < 0.001.

**Figure 5 ijms-23-04511-f005:**
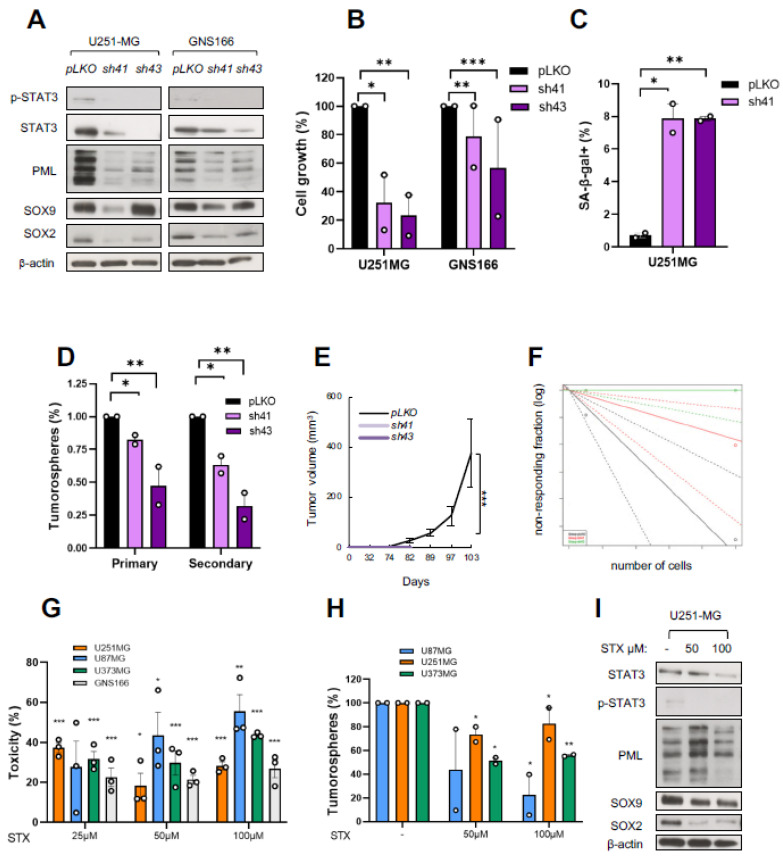
Genetic and pharmacological *STAT3* inhibition in glioma cells: (**A**) representative immunoblots of p-STAT3, STAT3, PML, SOX9, and SOX2 in U251-MG and GNS166 cells in STAT3 knockdown (“*sh41*” and “*sh43*”) and control (“*pLKO*”) cells (*n* = 3); (**B**) quantification of cell growth in *sh41* and *sh43* cells compared with controls (*n* = 2); (**C**) quantification of SA-β-gal^+^ cells in *STAT3* knockdown cells (*n* = 2); (**D**) quantification of tumorspheres (primary and secondary) in *sh41* and *sh43* U251-MG cells relative to *pLKO* condition (*n* = 2); (**E**) tumor volume representation at indicated time points after subcutaneous injection of U251-MG cells (*n* = 4); (**F**) ELDA plot of limiting dilution assay of *sh41, sh43*, and control U251-MG cells; (**G**) cytotoxicity exhibited by indicated glioma cells after treatment with increasing doses of STX-0119 for 72 h (*n* = 6); (**H**) quantification of tumorspheres in indicated cells after treatment with 50 and 100 µM STX-0119 (*n* = 3). Calculations were relative to untreated cells; (**I**) representative immunoblot of p-STAT3, STAT3, PML, SOX9, and SOX2 expression in U251-MG cells after treatment with 50 and 100 µM STX-0119 for 72 h (*n* = 3). * *p* < 0.05, ** *p* < 0.01, *** *p* < 0.001.

**Figure 6 ijms-23-04511-f006:**
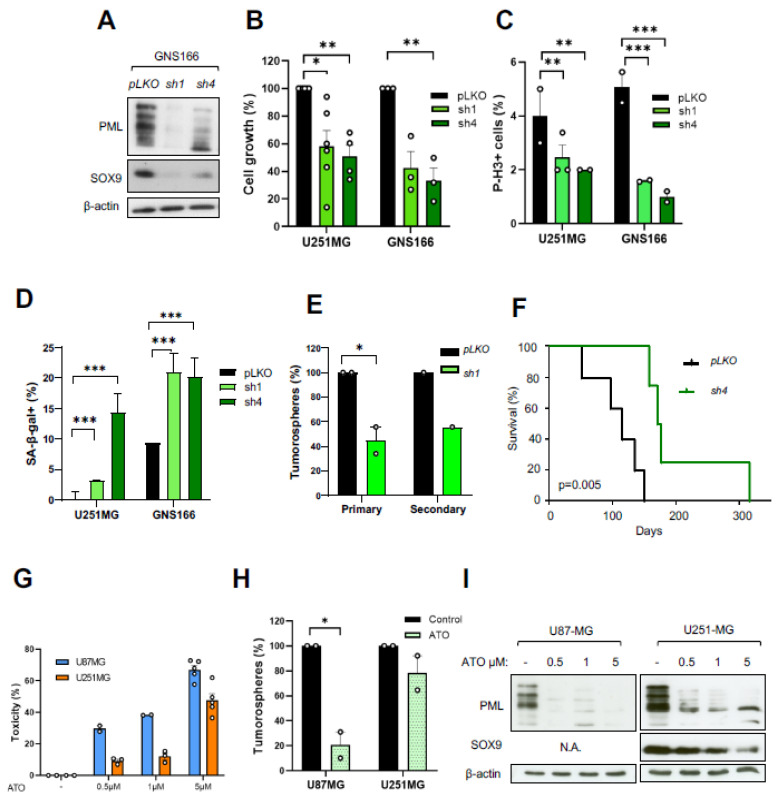
Genetic and pharmacological *PML* inhibition in glioma cells: (**A**) representative immunoblot of PML and SOX9 expression in *sh1* and *sh4 PML* knockdown and *pLKO* control GNS166 cells (*n* = 3); (**B**) quantification of cell growth in control and *PML* knockdown cells (*n* = 3); (**C**) quantification of P-H3^+^ in control and *PML* knockdown (*n* = 3); (**D**) quantification of SA-β-gal^+^ cells in control and *PML* knockdown (*n* = 3); (**E**) quantification of primary and secondary tumorspheres in *PML* knockdown U251-MG cells. Quantifications are relative to *pLKO* conditions; (**F**) Kaplan–Meier curves of mice after stereotactic injection of GNS166 control cells (*n* = 5) and *shPML-4* cells (*n* = 5); (**G**) cytotoxicity of indicated glioma cells after treatment with increasing doses of arsenic trioxide (ATO) for 48 h (*n* = 5). Calculations were relative to untreated cells; (**H**) relative quantification of oncospheres after treatment with 1 µM ATO (*n* = 2); (**I**) representative immunoblots of PML and SOX9 expression in glioma cells after treatment with 0.5, 1, and 5 µM ATO for 48 h (*n* = 3). * *p* < 0.05, ** *p* < 0.01, *** *p* < 0.001.

**Figure 7 ijms-23-04511-f007:**
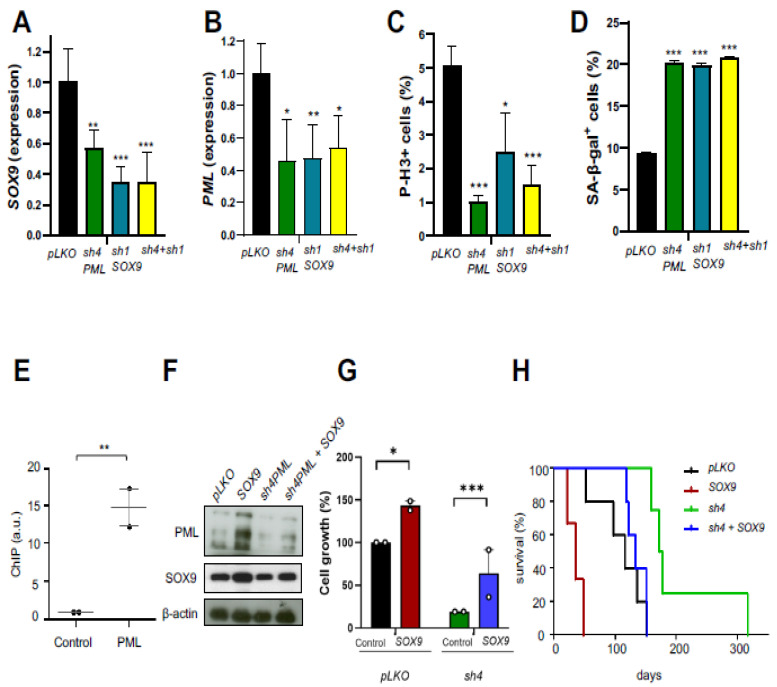
The SOX9–STAT3–PML axis constitutes a regulatory loop in GSCs function: (**A**,**B**) *SOX9* and *PML* mRNA expression in control (*pLKO*) cells, *PML* knockdown (*sh4*), *SOX9* knockdown (*sh1*) and *PML*-*SOX9* co-silenced (*sh4* + *sh1*) GNS166 cells (*n* = 3); (**C**) quantification of P-H3^+^ in indicated genotypes (*n* = 3); (**D**) quantification of SA-β-gal^+^ in indicated genotypes (*n* = 3); (**E**) representation of chromatin immunoprecipitation assay of ectopic PML in U251-MG cells after induction of 50 ng/mL doxycycline during 3 days. Data were normalized to IgG. a.u., arbitrary units. (*n* = 2). (**F**) representative immunoblot of PML and SOX9 protein levels in *pLKO* and *PML* knockdown (*sh4*) with or without overexpression of SOX9 in GNS166 cells (*n* = 3); (**G**) quantification of cell growth in indicated genotypes (*n* = 3); (**H**) Kaplan–Meier curves of mice stereotactically injected with GNS166 cells for the four conditions. * *p* < 0.05, ** *p* < 0.01, *** *p* < 0.001.

## Data Availability

The data that support the findings of this study are included in the manuscript and Appendix A. Microarray data are openly available in GEO database with access number GSE181035.

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
