# Peer review of "High SOX9 Maintains Glioma Stem Cell Activity through a Regulatory Loop Involving STAT3 and PML"

_ijms, 2022, doi:10.3390/ijms23094511_

Round 1

Reviewer 1 Report

In this work, the authors have elucidated a SOX9-STAT3-PML signaling axis to be a molecular mechanism regulating glioma stem cells, and glioma pathology. Although the data is well presented, the work lacks novelty, since there's extensive amounts of literature demonstrating the functional relevance of SOX9 in glioma biology, and highlighting different molecular mechanisms by which it does so. There are some obvious gaps in the work, which the authors should attempt to address, in order to increase the impact of their work.

  1. Is SOX9 required for oncogenic transformation of NSCs into GSCs?
  2. Does SOX9, or the SOX9-STAT3-PML axis show any trend of expression from lower to higher grade of glioma?
  3. Did the authors perform secondary transplants with SOX9 glioma cells? primary transplants suggest proliferative capacity, but a truer test of stemness would be secondary transplants.
  4. Can the authors show the efficacy of PML inhibition with ATO treatment, and SOX9 down-regulation, in patient samples? Would the two treatments be synergistic in effect?

Author Response

In this work, the authors have elucidated a SOX9-STAT3-PML signaling axis to be a molecular mechanism regulating glioma stem cells, and glioma pathology. Although the data is well presented, the work lacks novelty, since there's extensive amounts of literature demonstrating the functional relevance of SOX9 in glioma biology, and highlighting different molecular mechanisms by which it does so. There are some obvious gaps in the work, which the authors should attempt to address, in order to increase the impact of their work.

AUTHORS : We thank the reviewer for considering the data is well presented. We agree that there are some manuscripts linking SOX9 to glioma progression. However, we respectfully consider that this is the first one identifying the underlying molecular mechanisms in patient derived GSCs and showing the link of SOX9 with STAT3.

  1. Is SOX9 required for oncogenic transformation of NSCs into GSCs?

AUTHORS: This question has been addressed previously in the manuscript Swartling et al., Cancer Cell 2012. PMID: 22624711. This information is part of the revised manuscript (See lines 73 and 74).

  1. Does SOX9, or the SOX9-STAT3-PML axis show any trend of expression from lower to higher grade of glioma?

AUTHORS: This information has been included in the revised version of the manuscript as Figure 1A and Figure 4J.

  1. Did the authors perform secondary transplants with SOX9 glioma cells? primary transplants suggest proliferative capacity, but a truer test of stemness would be secondary transplants.

AUTHORS: We thank the reviewer for this insightful suggestion. We did not perform the secondary transplants in vivo but the performed the secondary tumorsphere assay in vitro. This information has been included in the revised version of the manuscript (See Fig. Suppl 1B). Moreover, and to further reinforce the link between SOX9 and stemness activity, we have included results of colony formation (See New Fig. 3E) and soft agar foci (See Fig. Suppl 1A) assays in control and SOX9 knockdown cells.

  1. Can the authors show the efficacy of PML inhibition with ATO treatment, and SOX9 down-regulation, in patient samples? Would the two treatments be synergistic in effect?

AUTHORS: This is an interesting question that can only be addressed by performing a clinical trial, which unfortunately is not ongoing at the moment. There are few clinical trials with ATO in glioma and we and others have previously shown that SOX9 levels are regulated by rapamycin or EGFR inhibitors. However, there are not currently ongoing clinical trials with any of those combinations.

Our preclinical results knocking down SOX9 and PML in combination (and in parallel with single knock down) reveal that they act in the same pathway and there is not synergistic effect limiting the possibility of success of the combined treatment.

Reviewer 2 Report

Aldaz et al have written a very nice manuscript on how high expression levels of Sox9/Stat3/Pml promote Glioma stem cell survival. Unfortunately high expression of these proteins also leads to poor patient survival. The results are well written and very clearly explained.

My only caveat is the low quality of figures. I don't know if it is editing process or something else but authors need to provide high resolution figures. Other than that, the paper is well written and clearly explains the groups findings.  

Author Response

Aldaz et al have written a very nice manuscript on how high expression levels of Sox9/Stat3/Pml promote Glioma stem cell survival. Unfortunately high expression of these proteins also leads to poor patient survival. The results are well written and very clearly explained.

My only caveat is the low quality of figures. I don't know if it is editing process or something else but authors need to provide high resolution figures. Other than that, the paper is well written and clearly explains the groups findings.  

AUTHORS: We apologize for the issue highlighted by the reviewer. We had reduced the quality of the images in order to maintain them in a reasonable size. However, following the suggestion of the reviewer, we increased them and we hope that this revised version is good enough for the reviewer and journal.

Reviewer 3 Report

The authors performed transcriptome analysis of SOX9 knockdown glioma cells and functional analysis of SOX9-related genes. They showed that SOX9, STAT3, and PML form a key regulatory loop for GSC activity and self-renewal.

This study is an extension of a series of authors’ studies related to the SOX family and may contribute to the elucidation of the positive correlation between SOX9, STAT3, and PML in glioblastoma.

  1. Page 4, 2.3: This section is the same as 2.1 and needs to be removed.
  2. Figures 3A, 4C, 5A, and 5I: The authors have shown reduced SOX2 expression in these figures. It is necessary to explain in the discussion how SOX2 is related to the SOX9-STAT3-PML axis.
  3. Figures 3D, 5C, 6D, and 7D: The authors have shown an increase in senescence as measured by cytoplasmic SA-beta-gal activity. Is the decrease in cell growth by SOX9 knockdown in Figure 3B due to senescence? Are cell differentiation and cell death involved in decreased cell growth?
  4. Figures 3E, 5D, and 6E: The authors showed the proportion of “tumorspheres” on the Y-axis, but the results in the text were shown by “oncosphere” formation. This inconsistency needs to be remedied.
  5. Page 6, transcriptomic analysis: In this analysis, the authors presented only the results of a cluster analysis. At least, results regarding SOX9, STAT3, and PML gene expression should be included in the text.
  6. Figure 4I: The authors need to explain which intensity grade is represented in the figures.
  7. Figure 4J: What does TMA mean? Does the result mean that high expression of SOX9 (SOX9++) is associated with elevated levels of PML? However, there seems to be no difference between SOX9+ and SOX9++ groups in PML protein level 3 (PML3).
  8. Figure 5G, page 10: The authors stated that “We found that increasing doses (25, 50, and 100 uM) of this drug significantly reduced viability in a dose-dependent manner in glioma and GSC cells”. No such tendency is observed in U251 and GNS166 cells. This needs to be rectified. Do cells treated with the STAT3 inhibitor fall into apoptosis or senescence?
  9. Figure 5I: Many bands can be seen in the PML lanes. Which band indicates the PML protein?
  10. Page 11, Figure 6: The title of Figure 6 is incorrect and “STAT3 inhibition” should be changed to “PML inhibition”.
  11. Page 13: Fig 7H is repeated in the text. The authors need to remove one of them.

Author Response

The authors performed transcriptome analysis of SOX9 knockdown glioma cells and functional analysis of SOX9-related genes. They showed that SOX9, STAT3, and PML form a key regulatory loop for GSC activity and self-renewal.

This study is an extension of a series of authors’ studies related to the SOX family and may contribute to the elucidation of the positive correlation between SOX9, STAT3, and PML in glioblastoma.

  1. Page 4, 2.3: This section is the same as 2.1 and needs to be removed.

AUTHORS: We apologize for the error that has been amended in the revised version of the manuscript.

  1. Figures 3A, 4C, 5A, and 5I: The authors have shown reduced SOX2 expression in these figures. It is necessary to explain in the discussion how SOX2 is related to the SOX9-STAT3-PML axis.

AUTHORS: This point has been discussed in the revised version of the manuscript.

  1. Figures 3D, 5C, 6D, and 7D: The authors have shown an increase in senescence as measured by cytoplasmic SA-beta-gal activity. Is the decrease in cell growth by SOX9 knockdown in Figure 3B due to senescence? Are cell differentiation and cell death involved in decreased cell growth?

AUTHORS: This is an interesting point that has been addressed and discussed in the revised version of the manuscript. We had previously described that SOX9 knock-down significantly increases cell death (Aldaz et al., Scientific Reports 2020) and this information has been incorporated in the text. Moreover, we have completed experiments that show that knock-down of SOX9 increases cell differentiation (New Fig 3F). These results are in line with previous results that correlate high levels of SOX9 with stem cell population and quiescence maintenance and overall confirm that SOX9 controls cell growth acting at multiple processes and pathways. 

  1. Figures 3E, 5D, and 6E: The authors showed the proportion of “tumorspheres” on the Y-axis, but the results in the text were shown by “oncosphere” formation. This inconsistency needs to be remedied.

AUTHORS: We thank the reviewer for this comment and the word oncosphere has been changed by tumorsphere within the text of the revised version of the manuscript.

  1. Page 6, transcriptomic analysis: In this analysis, the authors presented only the results of a cluster analysis. At least, results regarding SOX9, STAT3, and PML gene expression should be included in the text.

AUTHORS: This information has been presented in the text of the manuscript. Moreover, a list of the genes differentially expressed between control and shSOX9 cells is part of the revised version of the manuscript (See Suppl Table 1).

  1. Figure 4I: The authors need to explain which intensity grade is represented in the figures.

AUTHORS: Information regarding the figure has been extended in the revised version of the text of the manuscript.

  1. Figure 4J: What does TMA mean? Does the result mean that high expression of SOX9 (SOX9++) is associated with elevated levels of PML? However, there seems to be no difference between SOX9+ and SOX9++ groups in PML protein level 3 (PML3).

AUTHORS: TMA means tissue microarray and this information is part of the Material and methods of the revised version of the manuscript. Yes, the results show that there is a significant correlation between the groups of high expression of SOX9 with high PML and high STAT3. The statistical analysis does not have into account only the cases in group 3 of PML but all the groups and in that circumstances the analysis show statistical significance (P< 0,0001). Additional information has been included in the text and figure legend to try to clarify this point.  

  1. Figure 5G, page 10: The authors stated that “We found that increasing doses (25, 50, and 100 uM) of this drug significantly reduced viability in a dose-dependent manner in glioma and GSC cells”. No such tendency is observed in U251 and GNS166 cells. This needs to be rectified. Do cells treated with the STAT3 inhibitor fall into apoptosis or senescence?

AUTHORS: Reviewer is correct and we do not see a dose dependent effect in all the cell lines. Consequently, we have removed the sentence from the text to be totally accurate with the results. The MTT studies suggest that there is reduction in cell viability likely related to apoptosis more than senescence. We didn´t complete detailed experiments for this issue but previous studies support this idea (Ashizawa et al., 2013 Int J Oncol PMID:23612755).

  1. Figure 5I: Many bands can be seen in the PML lanes. Which band indicates the PML protein?

AUTHORS: Yes, this is a characteristic of PML protein and has been previously described by us (Martin-Martin et al., 2016 Nature Communications) and others in different types of cancer including glioblastoma (Iwanami et al., 2013 PNAS).

  1. Page 11, Figure 6: The title of Figure 6 is incorrect and “STAT3 inhibition” should be changed to “PML inhibition”.

AUTHORS: We apologize for the error, which has been corrected in the revised version of the manuscript.

  1. Page 13: Fig 7H is repeated in the text. The authors need to remove one of them.

AUTHORS: The excluded one of the Fig 7H as suggested by the reviewer.

Round 2

Reviewer 1 Report

The authors have addressed all comments.